# Chronic-Antibiotics Induced Gut Microbiota Dysbiosis Rescues Memory Impairment and Reduces β-Amyloid Aggregation in a Preclinical Alzheimer’s Disease Model

**DOI:** 10.3390/ijms23158209

**Published:** 2022-07-26

**Authors:** Paola C. Bello-Medina, Karina Corona-Cervantes, Norma Gabriela Zavala Torres, Antonio González, Marcel Pérez-Morales, Diego A. González-Franco, Astrid Gómez, Jaime García-Mena, Sofía Díaz-Cintra, Gustavo Pacheco-López

**Affiliations:** 1Departamento de Neurobiología del Desarrollo y Neurofisiología, Instituto de Neurobiología, Universidad Nacional Autónoma de México, Querétaro 76230, Mexico; p_bello@correo.ler.uam.mx; 2Biological and Health Sciences Division, Campus Lerma, Metropolitan Autonomus University (UAM), Lerma 52005, Mexico; a_gonzalezs@correo.ler.uam.mx (A.G.); m.perez@correo.ler.uam.mx (M.P.-M.); da_gonzalez@correo.ler.uam.mx (D.A.G.-F.); 2173073699@correo.ler.uam.mx (A.G.); g.pacheco@correo.ler.uam.mx (G.P.-L.); 3Departamento de Genética y Biología Molecular, Centro de Investigación y de Estudios Avanzados del Instituto Politécnico Nacional, Unidad Zacatenco, Mexico City 07360, Mexico; karina.corona@cinvestav.mx (K.C.-C.); norma.zavala@cinvestav.mx (N.G.Z.T.)

**Keywords:** dysbiosis, novel-object localization, firmicutes, bacteroidetes, alpha-diversity, beta-diversity, antibiotics, high-throughput DNA sequencing, fecal bacterial microbiota

## Abstract

Alzheimer’s disease (AD) is a multifactorial pathology characterized by β-amyloid (Aβ) deposits, Tau hyperphosphorylation, neuroinflammatory response, and cognitive deficit. Changes in the bacterial gut microbiota (BGM) have been reported as a possible etiological factor of AD. We assessed in offspring (F1) 3xTg, the effect of BGM dysbiosisdysbiosis in mothers (F0) at gestation and F1 from lactation up to the age of 5 months on Aβ and Tau levels in the hippocampus, as well as on spatial memory at the early symptomatic stage of AD. We found that BGM dysbiosisdysbiosis with antibiotics (Abx) treatment in F0 was vertically transferred to their F1 3xTg mice, as observed on postnatal day (PD) 30 and 150. On PD150, we observed a delay in spatial memory impairment and Aβ deposits, but not in Tau and pTau protein in the hippocampus at the early symptomatic stage of AD. These effects are correlated with relative abundance of bacteria and alpha diversity, and are specific to bacterial consortia. Our results suggest that this specific BGM could reduce neuroinflammatory responses related to cerebral amyloidosis and cognitive deficit and activate metabolic pathways associated with the biosynthesis of triggering or protective molecules for AD.

## 1. Introduction

Currently, more than 55 million people suffer from dementia worldwide, and there are nearly 10 million new cases every year. Alzheimer’s disease (AD) has been the most common dementia contributing to 60–70% of cases. AD is a neurodegenerative disorder characterized by β-amyloid peptide (Aβ) plaques and tau neurofibrillary tangle formation [1,2] with progressive cognitive deficit [1,3]. A model for studying AD is the triple transgenic mouse (3xTg-AD), which contains three mutations associated with familial AD (APP Swedish, MAPT P301L, and PSEN1 M146V mutations). In this model, extracellular Aβ accumulation within the hippocampus (HIP) appears when mice are 6 months old (mo), and tau changes at 12 to 15 mo, with hyperphosphorylated tau (pTau) aggregates also detected in the HIP [4]. These histological changes are associated with cognitive impairment in behavioral paradigms, such as the T-maze, elevated plus maze, Morris water maze (MWM), and object recognition [4,5,6,7,8,9]. However, few reports have studied alterations at the early symptomatic stage of AD (before six mo). In the early symptomatic stage of AD researchers observed Aβ accumulation and pTau related to changes in theta oscillations in the HIP; postsynaptic potential alterations in CA1 of the HIP; decreases in CA1 neuronal excitability; decreases in dendritic spine density; low release of dopamine, norepinephrine, and glutamate; and memory impairment in novel object localization (NOL), 8-arm radial maze, and MWM [10,11,12,13,14,15].

Several factors are linked to AD development and progression such as neuroinflammation, lipid and glucose metabolism alterations, oxidative stress, and synaptic plasticity impairment [6,16,17,18,19,20,21,22,23,24]. Recent studies have focused on the role of BGM dysbiosis. Dysbiosis refers to the loss of homeostasis between commensal, symbiotic, and pathogenic bacteria within the intestine functionally related to the host [25,26]. Thus, at the symptomatic stage (beyond 6 mo), an increase in Proteobacteria and Firmicutes and a decrease in Bacteroidetes phyla in 5xFAD mice have been observed, whereas an increase in Bacteroidetes and Tenericutes and a decrease in Actinobacteria, Proteobacteria, and Firmicutes phyla have been reported in APPPS1 mice. Regarding bacterial abundance, Rikenellaceae increases, and both *Akkermansia* and *Allobaculum* genera decrease in these transgenic models [27,28]. In 3xTg mice, an increase in Firmicutes and Bacteroidetes and a decrease in Proteobacteria, Cyanobacteria, Verrucomicrobia, and Tenericutes were observed [27,28,29]. On the other hand, in the early symptomatic stage of AD, an increase in Firmicutes and a decrease in Bacteroidetes, Actinobacteria, and TM7 phyla have been reported [10,30,31].

BGM is composed of gram-negative and -positive bacteria, i.e., Bacteroidetes and Firmicutes, respectively. A component of the extracellular membrane of gram-negative bacteria is lipopolysaccharides (LPS) [32] which travel via bloodstream from the gut lumen to the cerebral parenchyma. LPS in cerebral structures induces a microglial and inflammatory response related to Aβ monomers, dimers, and oligomers; this suggests that the components and metabolites produced for the BGM could participate in amyloid neurotoxicity and could contribute as etiological factors of AD [33,34,35,36].

It is unknown if specific BGM members could act as predisposing factors for cognitive deficits and the Aβ and tau accumulation that characterize AD. In this study, we analyzed the composition changes of BGM induced by chronic antibiotic (Abx) treatment, transmitted from mother to offspring. Furthermore, we investigated if these changes attenuate or delay the spatial memory impairment, total Aβ and Aβ1-42 accumulation, and total tau and phosphorylated tau in the HIP, at the early symptomatic stage in the 3xTg preclinical model of AD.

## 2. Results

### 2.1. A Habituation Period for Abx Consumption Was Observed in Offspring after Weaning

The F1 NoTg-Ctrl and 3xTg-Ctrl mice maintained a stable fluid intake from weaning on PD30 until PD60. While F1 NoTg-Abx showed a decrease in liquid intake from days 1 to 8, this effect was also observed in the 3xTg-Abx group. A decrease in fluid consumption for 14 days was observed in 3xTg-Abx (Appendix A). Since then, NoTg-Abx and 3xTg-Abx equaled the fluid consumption of NoTg-Ctrl and 3xTg-Ctrl, showing no Abx avoidance.

### 2.2. Offspring Body Weight Was Affected by Abx Consumption

Body weight was higher in NoTg-Ctrl mice in comparison to NoTg-Abx mice at 3 to 9 (*p* < 0.05 to < 0.0001), 11 (*p* < 0.05), 14 (*p* < 0.05), 15 (*p* < 0.01), and 17 to 23 (*p* < 0.01 to < 0.0001) weeks of age (Appendix A); the same effect was observed in 3xTg-Abx compared to 3xTg-Ctrl mice at 3 to 10 (*p* < 0.01 to < 0.0001), 15 (*p* < 0.05), and 17 to 23 (*p* < 0.05 to < 0.0001) weeks of age (Appendix A).

### 2.3. BGM Perturbation with Abx Produces a Delayed Motor Alteration in 3xTg

During the NOL habituation session, there were significant differences in resting time. The Bonferroni post hoc test showed a decrease in resting time in 3xTg-CtrlCtrl in comparison to NoTg-Ctrl (*p* = 0.001) and NoTg-Abx (*p* = 0.005). No differences were observed in speed or traveled distance (Figure 1a,b). These results demonstrate that 3xTg-CtrlCtrl mice had a subtle motor alteration on PD150 (Figure 1c).

### 2.4. BGM Dysbiosis with Abx Attenuates NOL Memory Impairment in 3xTg Mice

A significant effect was found in the object exploration time during training between groups. Exploration time was lower in Fam1 than in Fam2 in 3xTg-Ctrl mice (*p* = 0.04) (Figure 1d). The exploration times for Fam1 and Fam2 were not significantly different from each other; this result reflects a good familiarization process, which is relevant for recognition memory acquisition [37]. To obtain a single measurement that represents training in the NOL and is related to BGM, we analyzed the preference percentage of familiar object localization and found no significant differences (Figure 1e). In the retention test, significant differences were observed in the exploration time of the novel object localization (Nov). Exploration time for Nov was higher than for the familiar object localization (Fam) in NoTg-Ctrl (*p* = 0.0006), NoTg-Abx (*p* < 0.0001), and 3xTg-Abx (*p* = 0.0081) mice (Figure 1f). No differences were found in exploration time between Nov and Fam in 3xTg-Ctrl; this result demonstrates that as expected there is a NOL memory deficit in 3xTg-Ctrl mice. Additionally, statistical differences were identified in Nov exploration time between NoTg-Ctrl and 3xTg-Ctrl groups (*p =* 0.0155) (Figure 1f). These results indicate a preference for the displaced object, which is the normal behavior for memory recognition [38]. Interestingly, Nov preference percentage was higher in 3xTg-Abx group in comparison to the 3xTg-Ctrl group, with no effects for NoTg-Ctrl and NoTg-Abx groups (*p* < 0.0001 for all comparisons) (Figure 1g), indicating a therapeutic effect of Abx treatment.

### 2.5. BGM Dysbiosis with Abx Treatment Attenuates Total Aβ in the Hippocampus of 3xTg Mice

There were significant differences in the total Aβ area ratio (BAM-10) in the *subiculum*, CA1, and CA3. The area ratio of BAM-10 was larger in 3xTg-Ctrl than in NoTg-Ctrl, NoTg-Abx, and 3xTg-Abx groups (*p* < 0.01 to <0.0001) (Figure 2a,b). Specifically, in the CA1, the BAM-10 area ratio was larger in 3xTg-Ctrl than in NoTg-Ctrl, NoTg-Abx, and 3xTg-Abx mice (*p* < 0.05 to <0.0001) (Figure 3a,c). In the CA3, the BAM-10 area ratio was larger in 3xTg-Ctrl mice than in NoTg-Ctrl, NoTg-Abx, and 3xTg-Abx mice (*p* < 0.05 to 0.0001) (Figure 2a,d), confirming now histologically the therapeutic effects of the Abx treatment.

### 2.6. BGM DysbiosisDysbiosis with Abx Treatment Alleviates Aβ1-42 in the Hippocampus of 3xTg Mice

In the *subiculum*, significant differences were reported in the Aβ1-42 area ratio. The Aβ1-42 area ratio was larger in 3xTg-Ctrl than in NoTg-Ctrl, NoTg-Abx, and 3xTg-Abx mice (*p* < 0.05 to 0.0001) (Figure 3a,b). In the CA1, the Aβ1-42 area ratio was larger in 3xTg-Ctrl than in NoTg-Ctrl, NoTg-Abx, and 3xTg-Abx groups (*p* < 0.05 to <0.0001) (Figure 3a,c). In the CA3, the Aβ1-42 area ratio was larger in 3xTg-Ctrl than in NoTg-Ctrl, NoTg-Abx, and 3xTg-Abx groups (*p* < 0.05 to <0.0001) (Figure 3a,d) again confirming histologically the therapeutic effects of the Abx treatment.

### 2.7. Total and Phosphorylated Tau Protein in the Hippocampus of 3xTg Mice Are Not Affected by BGM DysbiosisDysbiosis with Antibiotics

In the *subiculum*, significant differences were found in the total Tau protein area ratio (Tau499). The area ratio of Tau499 was larger in 3xTg-Ctrl and 3xTg-Abx than in NoTg-Ctrl and NoTg-Abx groups (*p* < 0.05 and 0.01, respectively) (Figure 4a). In the CA1, the Bonferroni test showed that the Tau499 area ratio was larger in 3xTg-Ctrl and 3xTg-Abx than in NoTg-Ctrl and NoTg-Abx mice (*p* < 0.05 and 0.01, respectively) (Figure 4b). In the CA3, no differences were observed (Figure 4c). In the *subiculum*, a significant effect was observed in the phosphorylated Tau area ratio (pTau) on Thr231. The Bonferroni post-hoc test showed that the pTau area ratio was larger in 3xTg-Ctrl and 3xTg-Abx than in NoTg-Ctrl and NoTg-Abx mice (*p* < 0.05 in all comparisons) (Figure 4d). No differences were identified in CA1 or CA3 (Figure 4e,f).

### 2.8. Chronic Abx Treatment Promotes Proliferation of Proteobacteria

The relative abundance of the BGM showed statistically significant changes (Appendix A). Specifically, in 3xTg-Abx vs 3xTg-Ctrl groups, we observed an increase in Proteobacteria and a decrease in Bacteroidetes and Firmicutes in F0 at GD18–19, and in their F1 at PD30 and PD150 (*p* < 0.05 to 0.0001) (Appendix A). However, F0 NoTg-Ctrl vs NoTg-Abx at GD18–19 comparisons did not show significant changes at phylum level, but their F1 NoTg-Abx showed a significant increase in Proteobacteria and decrease in Bacteroidetes and Firmicutes at PD30 and PD150 (*p* < 0.001 to 0.0001) (Appendix A).

For the bacterial alpha-diversity indexes, statistically significant changes were shown for all comparisons (Appendix A). In F0 at GD18–19, significant differences were observed in NoTg-Ctrl vs NoTg-Abx (Chao1 *p* = 0.014, Shannon *p* = 0.014, and Simpson *p* = 0.016) and 3xTg-Ctrl vs 3xTg-Abx (Chao1 *p* = 0.014, Shannon *p* = 0.014, and Simpson *p* = 0.016) (Appendix A). In F1 at PD30, a significant effect was observed in NoTg-Ctrl vs NoTg-Abx (Chao1 *p* = 0.001, Shannon *p* = 0.001, and Simpson *p* = 0.001) and 3xTg-Ctrl vs 3xTg-Abx (Chao1 *p* = 0.001, Shannon *p* = 0.001, and Simpson *p* = 0.001) (Appendix A). In F1 at PD150, significant differences were observed in NoTg-Ctrl vs NoTg-Abx (Chao1 *p* = 0.001, Shannon *p* = 0.001, and Simpson *p* = 0.001) and 3xTg-Ctrl vs 3xTg-Abx (Chao1 *p* = 0.001, Shannon *p* = 0.042, and Simpson *p* = 0.012) (Appendix A).

For the beta-diversity analysis, we found that the Abx treatment significantly influenced the bacterial community in all comparisons. We observed a clustering separating both groups related to Ctrl and Abx treatments in F0 NoTg (*p* = 0.013) and F0 3xTg (*p* = 0.002) at GD18–19 (Appendix A); in F1 NoTg (*p* = 0.001) and F1 3xTg at PD30 (*p* = 0.001) (Appendix A); and in F1 NoTg (*p* = 0.001) and F1 3xTg (*p* = 0.011) at PD150 (Appendix A).

### 2.9. A Vertical Transfer of BGM from F0 to F1 Was Observed at PD30

Firmicutes (5–28%) and Bacteroidetes (33–75%) are predominantly observed in parents with no antibiotics treatment (a, F0 chart; NoTg-Ctrl and 3xTg-Ctrl). Both phyla were vertically transferred to the offspring (F1 NoTg-Ctrl and F1 NoTg-Ctrl). The microbiome is predominantly made up of the Proteobacteria phylum; a vertical transfer preserving a similar relative abundance percentage of the Proteobacteria phylum (89–94%) was also observed (Figure 5).

### 2.10. Bacterial Diversity Is Directly Correlated with NOL Performance and Total Aβ and Aβ1-42 in the Hippocampus

The Shannon diversity index of F1 3xTg-Ctrl and 3xTg-Abx mice at PD150 positively correlates with NOL data (Appendix A) and the total Aβ and Aβ1-42 area ratio in the *subiculum*, CA1, and CA3 of the HIP (Appendix A). The preference percentage of familiar object localization (NOL training) was not correlated with bacterial diversity (*p* = 0.0633) (Appendix A). However, a statistically significant correlation was observed between the Shannon diversity index and the preference percentage of the NOL retention test in the F1 3xTg-Ctrl and 3xTg-Abx groups at PD150 (*p* = 0.0113) (Appendix A). Otherwise, positive correlations were found between the total Aβ (BAM-10) area ratio in the *subiculum*, CA1, or CA3, and the Shannon diversity index (*p* = 0.003, 0.0007, and 0.0012, respectively) (Appendix A). Moreover, there was a positive correlation between the Shannon diversity index and the Aβ1-42 area ratio in the *subiculum*, CA1, or CA3 in F1 3xTg-Ctrl and 3xTg-Abx mice at PD150 (*p* = 0.072, 0.008, and 0.017, respectively) (Appendix A). These results indicate that F1 3xTg-Abx mice at PD150 had a low Shannon index value and a low total Aβ and Aβ1-42 area ratio in the *subiculum*, CA1, and CA3 of the HIP. Likewise, F1 3xTg-Ctrl at PD150 presented a large total Aβ and Aβ1-42 area ratio in the *subiculum*, CA1, and CA3 and a high Shannon diversity index (Appendix A).

### 2.11. NOL Performance and Aβ in the Hippocampus Correlates with BGM Taxa in 3xTg Mice at PD150

To investigate whether the BGM is implicated in spatial memory impairment and the accumulation of total Aβ and Aβ1-42 in HIP triggered by AD, we performed Spearman correlation analyses using BGM abundance data, NOL data, and immunohistochemistry data of the total Aβ and Aβ1-42 area ratio in the *subiculum*, CA1, and CA3 in F1 3xTg mice at PD150 (all *p*-values are listed in Appendix A) and observed interesting tendencies. In the NOL paradigm, we found that the preference percentage of familiar object localization (training) suggest positive association with the Lachnospiraceae family and the bacterial genera *Clostridium*, *Lactobacillus*, *Bifidobacterium*, *Allobaculum*, AF12, *Ruminococcus*, *Mucispirillum*, *Streptococcus*, and negatively associated with genera *Mycoplasma* and *Sutterella* (Figure 6). In the preference percentage of novel object localization (retention test), we observed a tendency of association with bacterial Desulfovibrionaceae, Lachnospiraceae, F16, Rikenellaceae, Coriobacteriaceae families; *Clostridium*, *Turicibacter*, *Dorea*, *Lactobacillus*, *Ruminococcus*, *Streptococcus genera*, and a negative association with *Mycoplasma*, *Sutterella*, *Klebsiella,* and *Enterobacter* genera (Figure 6). In motor activity parameters evaluated during the habituation session of NOL, resting time was positively associated with Ruminococcaceae, Rikenellaceae, F16, and Muribaculaceae (S24-7) families (Figure 6). In addition, speed and traveled distance were associated positively in the *Streptococcus* genus and negatively with the Muribaculaceae family (Figure 6). 

Concerning the bacterial taxa correlated with the Aβ1-42 area ratio in the HIP of F1 3xTg mice at PD150, the Lachnospiraceae, Desulfovibrionaceae, Ruminococcaceae, Rikenellaceae families and AF12, *Clostridium*, *Prevotella*, *Streptococcus*, *Paraprevotella*, *Parabacteroides*, *Bacteroides* genera were positively associated with the total Aβ area ratio found in the *subiculum*, and negatively associated with Enterobacteriaceae family, and *Klebsiella*, *Enterobacter*, *Mycoplasma* and *Sutterella* genera (Figure 7).

The Aβ1-42 area ratio in the CA3 has a tendency of positive correlation with the Clostridiales and Bacteroidales order; Lachnospiraceae, Desulfovibrionaceae, Rikenellaceae, Muribaculaceae, Enterobacteriaceae families; and the AF12, *Clostridium*, *Oscillospira*, *Prevotella*, and *Lactobacillus* genera (Figure 7), and negatively correlated with the Enterobacteriaceae family and the *Klebsiella*, *Enterobacter*, and *Mycoplasma* genera (Figure 7). Similarly, the Aβ1-42 area ratio in the CA1 brain was positively correlated with the Ruminococcaceae, Rikenellaceae, and F16 families and negatively correlated with the *Mycoplasma* genus (Figure 7). 

The total Aβ (BAM-10) area ratio in the *subiculum* was positively associated with Lachnospiraceae and Rikenellaceae families and AF12, *Ruminococcus*, *Dorea*, *Anaeroplasma*, *Turicibacter*, and *Lactobacillus* genera. In addition, it is negatively associated with the Enterobacteriaceae family and the *Mycoplasma* and *Sutterella* genera (Figure 7). Regarding the CA3, the total Aβ area ratio was positively correlated with the Rikenellaceae, Muribaculaceae, and F16 families and the *Oridobacter* and *Lactobacillus* genera (Figure 7), and negatively correlated with the Enterobacteriaceae family and *Klebsiella*, *Mycoplasma*, and *Sutterella* genera (Figure 7). The total Aβ (BAM10) area ratio in the CA1 was only positively correlated with the Rikenellaceae family and negatively correlated with the *Sutterella* genus (Figure 7).

### 2.12. There Is a Differential Abundance in the BGM Composition in All Groups

Linear discriminant analysis (LEfSe) was used to identify the differential abundance of bacteria, using an LDA Score of 2.5 as cutoff value. Comparing all groups in F0 at GD18–19. In Ctrl groups, F0 NoTg showed an increase in the Clostridia order, the Muribaculaceae family and the *Coprococcus* and *Lactobacillus* genera. In addition, 3xTg mice exhibited an increase in the *Prevotella* genus. However, in the Abx groups, NoTg only showed an increase in the *Enterobacter* genus, and 3xTg exhibited an increase in the *Klebsiella* and *Sutterella* genera (Appendix A).

For F1 NoTg-Ctrl, NoTg-Abx, 3xTg-Ctrl, and 3xTg-Abx at PD30 comparisons, the NoTg-Abx group showed an increase in the Enterobacteriaceae family, and the 3xTg-Ctrl group presented an increase in the Lachnospiraceae family (Appendix A).

Concerning the differential abundance of bacteria inF1 NoTg-Ctrl, NoTg-Abx, 3xTg-Ctrl, and 3xTg-Abx mice at PD150, we observed in NoTg-Ctrl an increase in the Clostridia order; in the Lachnospiraceae, Rikenellaceae, and F-16 families; and in the *Lactobacillus*, *Prevotella*, *Ruminococcus*, *Clostridium*, and *Odoribacter* genera. The 3xTg-Ctrl group exhibited an increase in the Erysipelotrichaceae, Ruminococcaceae, Helicobacteraceae, and Muribaculaceae families; in the *Prevotella*, *Bacteroidetes*, *Oscillospira,* and *AF12* genera. Moreover, NoTg-Abx mice showed a differential bacterial abundance with an increase in the Enterobacteriaceae family and in the *Enterobacter*, *Sutterella*, and *Klebsiella* genera (Appendix A). 

A different profile in the differential abundances of bacteria was observed in Ctrl and Abx in F0 NoTg and 3xTg mice at GD18–19 and F1 mice at PD30 and PD150 (Appendix A).

### 2.13. The Predicted Functional Metagenome Shows Reduced Detonator Molecules and Greater Metabolism of Protective Molecules after Abx Treatment

PICRUSt analysis shows a low pyruvate fermentation to acetone, lysine fermentation to acetate and butanoate, pyruvate fermentation to propanoate, CDP-diacylglycerol biosynthesis, and peptidoglycan biosynthesis metabolism in NoTg-Abx and 3xTg-Abx mice in comparison with NoTg-Ctrl and 3xTg-Ctrl mice, respectively. Interestingly, AcetilCoA fermentation to butanoate was identified in 3xTg-Abx compared to 3xTg-Ctrl mice (*p*-values *<* 0.05 to <0.001 for each comparison) (Figure 8a–f); these metabolic pathways participate in AD appearance and progression such as short-chain fatty acids (SCFA: acetate, propionate, and butyrate), diacylglycerol related to apolipoprotein E, and peptidoglycan. Oppositely, the dysbiosis in BGM with Abx treatment produced an increase in 1,4-dihydroxy-2-naphthoate, 8-amino-7-oxononanoate, pyridoxal 5′-phosphate, biotin, oleate, L-alanine, L-phenylalanine, and L-tyrosine (*p*-values < 0.05 to <0.001 for each comparison) biosynthesis in NoTg-Abx and 3xTg-Abx groups in comparison with NoTg-Ctrl and 3xTg-Ctrl groups, respectively (Figure 8g–n); these metabolic pathways are related to protective AD such as vitamins and omega fatty acids biosynthesis, and neurotransmitter precursors. All these effects on functional metagenomics occur specifically in the mice that received Abx treatment regardless of the genotype.

## 3. Discussion

We studied vertical transference from F0 to F1 NoTg and 3xTg mice, and our results suggest that BGM composition changes due to an Abx treatment in F0 are transmitted to their F1. We reported that the BGM dysbiosis characteristic of AD was modified in 3xTg-Abx mice. Importantly, we found that a BGM dysbiosis with Abx treatment delays or attenuates spatial memory impairment and total Aβ and Aβ 1-42, but not in total Tau and phosphorylated Tau in the *subiculum*, CA1, and CA3 at an early symptomatic stage of AD. On the other hand, we observed that alpha diversity correlates with NOL retention performance, total Aβ and Aβ1-42 accumulation, and with specific bacterial consortia. 

We found that F1 NoTg-Abx and 3xTg-Abx mice decrease Abx consumption for 9 and 15 days, respectively. These results are due to the neophobia that occurs in mice that are exposed to a novel taste experience. Previous studies show that 3xTg mice exhibit habituation impairment related to a cholinergic deficit and alteration in perirhinal cortex function, both associated with the progression of AD [39,40,41,42]. Furthermore, we observed a body weight decrease in NoTg-Abx and 3xTg-Abx groups; this result agrees with the outcome obtained by Reikvam et al., who reported a decrease in body weight and liquid consumption after Abx cocktail administration in drinking water related to the foul taste of metronidazole [43].

Interestingly, we reported that BGM composition in F0 NoTg-Ctrl and 3xTg-Ctrl mice and the BGM dysbiosis in F0 NoTg-Abx and 3xTg-Abx mice were transmitted to their F1. Specifically, in the F0 3xTg-Abx group, an increase in Proteobacteria and a decrease in Bacteroidetes and Firmicutes phyla were observed; the same relative abundance was detected in F1 mice at PD30 and PD150. In the 3xTg-Abx group, we found an increase in the Actinobacteria phylum in comparison to the 3xTg-Ctrl group. It is important to notice that Actinobacteria have been associated with maintaining gut barrier homeostasis, a decrease in intestinal permeability, and anti-inflammatory response [44,45]. Further, a decrease in Actinobacteria is related to aging [46].

In F1 NoTg-Abx and 3xTg-Abx mice, we observed an increase in Proteobacteria and a decrease in Bacteroidetes and Firmicutes phyla; these effects were also previously reported in NoTg and AD transgenic mouse models [47,48,49,50]. Recent studies have shown a decrease in Proteobacteria, and have related increased levels of Bacteroidetes and Firmicutes to AD and aging [10,28,29,44,46,51,52]. The vertical transference from F0 to their F1 could be occurring through vaginal microbiota [53,54] during birth, milk microbiota [55] during lactation, and coprophagia that is common in rodents [56]. Vertical transference of BGM suggests that the dysbiosis observed in AD mice or patients could be transmitted to their offspring enhancing the amount of the factors that trigger the onset of AD in early adulthood.

Concerning alpha diversity, we found a decrease in the Simpson and Chao1 index in F0 and F1 NoTg-Abx and 3xTg-Abx groups; a similar decrease was also observed after chronic Abx treatment in NoTg and 5xFAD mice at 4 mo [57].

Regarding beta-diversity analysis, we observed a clustering separating both groups related to Ctrl and Abx treatments in NoTg and 3xTg in F0 at GD18–19 and F1 at PD30 and PD150. The same effect was reported previously in adult AD transgenic [57,58] and NoTg [50] mice.

In the LEfSe analysis for BGM, we reported differential dynamics of BGM associated with each life stage such as gestation (GD18–19), childhood (PD30), and early adulthood (PD150) and with treatment and genotype. In F0 NoTg-Ctrl and 3xTg-Ctrl groups at GD18–19, we observed an increase in Prevotellaceae and Paraprevotellaceae families, while in F0 3xTg mice we observed an increase in Paraprevotella and Bacteroides genera. Moreover, in the F0 NoTg-Ctrl and 3xTg-Ctrl groups at GD18–19 and F1 at PD30, we found an increase in the Muribaculaceae family and *Parabacteroides* genus; these bacterial consortia have been reported in fetal lung and placental humans during pregnancy [59], confirming BGM vertical transference. In NoTg-Ctrl mice at PD150, the *Erysipelotrichaceae* and Rikenellaceae families, and *Odirobacter* and *Clostridium* genera were identified; these results agree with previous reports in NoTg mice [10,28]. However, in 3xTg-Ctrl at 5 mo, an increase was found in Helicobacteraceae and Desulfovibrionaceae families and *Bifidobacterium, Dorea*, and *Allobaculum* genera [10].

Furthermore, we show BGM changes related to AD progression from gestation to early adulthood. In F0 3xTg-Ctrl mice at GD18–19, an increase in the Rikenellaceae family and in the *Clostridium*, *Odoribacter*, *Dehalobacterium* genera was reported. In F1 at PD30, we found an increase in Helicobacteraceae, Erysipelotrichaceae, and Desulfovibrionaceae families, and in *Odoribacter*, *Streptococcus*, and *Coprobacillus* genera. In F1 at PD150, we found an increase in Helicobacteraceae, Desulfovibrionaceae, and Muribaculaceae families, and in *Bifidobacterium*, *Dorea*, *Paraprevotella*, *Allobaculum* genera. These BGM consortia alterations could be related to vertical transference and AD progression. Likewise, the antibiotics treatment also produced in NoTg and 3xTg an increase in the Enterobacteriaceae family and in the *Enterobacter*, *Klebsiella*, and *Sutterella* genera in F0 at GD18–19 and in F1 at PD30 and PD150; these BGM changes were reported after Abx treatment [47,49,50].

In this study, we observed motor activity alterations in 3xTg-Ctrl mice at PD150. Previous studies in 3xTg mice have demonstrated that this alteration is produced in the later symptomatic stage of AD [60,61]. In the cognitive aspect, we found a subtle NOL learning deficit in the exploration time variable; however, in preference percentage, this was not observed because the preference percentage is a proportion that can dilute the subtle effect observed in 3xTg at 5 mo [10,38]. On the other hand, in the NOL retention test, we found a spatial memory impairment in 3xTg-Ctrl mice, as previously reported [10,11]. 

Here we reported an increase in total Aβ and Aβ1-42 accumulation in the *subiculum*, CA1, and CA3 of the HIP from 3xTg-Ctrl mice, as previously reported [10,11,12]. We also observed an increase in total Tau protein in the *subiculum* and CA1 as well as an increase in phosphorylated Tau in the *subiculum* in 3xTg-Ctrl and 3xTg-Abx mice; these results have been reported previously [62]. However, total Tau and pTau dynamics were circumscribed to these hippocampal areas because Tau changes were observed in 3xTg mice at the late symptomatic stage of AD [4,62,63,64]. 

Surprisingly, we found an amelioration in motor activity alteration, NOL training, retention test impairment, and improvement in total Aβ and Aβ1-42 accumulation in the *subiculum*, CA1, and CA3 of 3xTg-Abx mice at PD150 when BGM was perturbed. These behavioral, cognitive, and histological effects are positively correlated with BGM alpha diversity, relative abundance, and specific bacterial consortia, demonstrating that specific bacteria could be related to the appearance of cognitive impairment and Aβ accumulation. In addition, the BGM dysbiosis has no significant effect on total Aβ or phosphorylated Tau nor does it have any correlation with BGM analysis.

It is well known that the BGM participates in the SCFA and vitamin synthesis and in the nervous and immune system [65,66,67]. In our study, functional metagenomic prediction analysis shows an interesting effect: a decrease in metabolic pathways associated with AD, triggering molecules such as SCFA, diacylglycerol, and peptidoglycan synthesis in NoTg-Abx and 3xTg-Abx groups. However, these effects are independent of transgenes in the 3xTg mouse model, but the improvement in NOL memory and delay in total Aβ and Aβ1-42 is specific to the presence of transgenes in 3xTg mice. Currently, it is known that SCFAs participate in the appearance and progression of AD. There are controversial data on the role of SCFAs in AD. SCFAs, such as acetate, propionate, and butyrate, are metabolic products of anaerobic fermentation by BGM [68] (e.g., Bacteroides, Lachnospiraceae, Ruminococcaceae, Rikenellaceae, Desulfovibrionaceae families) [69]. There is evidence that acetate participates in the pro-inflammatory response and that propionate and butyrate contribute to the anti-inflammatory response, specifically regulating the levels of TNFα, and to the activation of TNF_K_B [70]. Other reports describe SCFAs as a fundamental physiopathological key that compromises the integrity of the tissue-blood barrier linking AD and the microbiota [71]. In addition, a decrease in acetate, propionate, and butyrate levels in plasma were observed in germ-free (GF) APPPS1 transgenic mice. These results are associated with an increase in Barnes maze performance. Furthermore, oral administration of SCFAs to GF APPPS1 mice produces an increase in the percentage of cerebral Aβ plaques, microglial activation phenotype, and microglial apolipoprotein E area [72]. There is evidence that acetate, propanoate, and butyrate participate in the inflammatory response related to amyloidogenesis and microglial response. MacFabe et al. [73] reported a social behavior alteration and an increase in oxidative stress and innate neuroinflammatory response produced by SCFA in patients with autism spectrum disorder. Likewise, AD patients with mild cognitive impairment exhibit high plasmatic and frontal cortex diacylglycerol levels in the early stage of AD [74].

BGM dysbiosis could lead to an increase in the growth of pathogenic bacteria that decrease intestinal barrier integrity, allowing pro-inflammatory molecules to travel throughout the bloodstream to the brain [75]. The most common gram-negative bacteria in the gastrointestinal tract are *Bacillus*, *Pseudomonas*, *Staphylococcus*, *Streptomyces*, and others that could be associated with amyloid deposits [36]. LPS and peptidoglycan are a component of gram-negative bacteria. These molecules are considered potent inflammatory and amyloidogenic factors in the parenchyma and around vessels in AD brains [76,77]. In addition, it is possible that BGM produces Aβ aggregates and insoluble proteins exhibiting β-pleated sheet structures, both of which induce Aβ plaque formation and enhance the risk for AD development [58,78]. Harach et al. (2017) [28] demonstrated that a gradual increase in BGM in aged APPPS1 mice is related to an increase in cerebral Aβ. 

In PICRUSt prediction analysis, we found an increase in metabolic pathways of protective metabolites for AD such as vitamin B (precursor, cofactor biotin, and vitamin B) and omega 9 biosynthesis, as well as neurotransmitter precursors such as alanine, phenylalanine, and tyrosine after BGM dysbiosis. *Bacteroides*, *Prevotella*, and *Lactobacillus* bacteria produce vitamin B that participates in immune system regulation [79]. Biotin or its precursors act as anti-inflammatory molecules, recovering the activity of the soluble guanylate cyclase, which is inhibited by Aβ [80]. Furthermore, researchers using the TgCRND8 transgenic model for AD observed that chronic administration of oleate (omega 9) induces a decrease in Aβ plaques in the cortex, HIP, and amygdala [81] and blocks inflammatory signaling in neuronal cultures [82].

In our study, we found negative correlations between Aβ in the HIP of 3xTg-Abx and some BGM such as *Klebsiella*, *Enterobacter*, *Mycoplasma*, and *Sutterella*. Previous studies reported the participation of these bacteria in neurotransmitters and their precursor’s biosynthesis such as dopamine, serotonin, GABA, norepinephrine, and histamine [83,84]. This finding is associated with our results reported in functional metagenomic prediction observed after Abx treatment. Taken together, these results suggest a recovery of neurotransmitter biosynthesis in the HIP of 3xTg mice at 5 mo [15].

Recently, BGM alterations have been reported with different strategies. The most relevant results show that GF APP transgenic mouse recolonization increased cerebral AD pathology [28]. In the APP-PS1 transgenic model, researchers reported changes in gut microbiota after Abx modified neuro-inflammatory responses and altered microglial morphology [58]. Probiotics administration in 3xTg mice decreases pro-inflammatory molecules such as IL-1β, IFN-γ, and TNF-α and cerebral Aβ [85].

Overall, BGM dysbiosis, such as relative abundance, alpha and beta diversity, and LEfSe analysis changes, caused in F0 3xTg mice at GD18–19 is vertically transferred to their F1 observed at PD30 and PD150. Our study suggests that BGM dysbiosis with Abx is closely linked to the delay in Aβ deposits in the HIP and cognitive deficit in the 3xTg preclinical model of AD at the early symptomatic stage of the pathology. Abx treatment modified BGM dysbiosis, allowing *Klebsiella*, *Enterobacter*, *Mycoplasma*, and *Sutterella* growth, which could induce a decrease in neuroinflammatory response related to cerebral amyloidosis. In addition, these bacterial consortia could be activating metabolic pathways related to the biosynthesis of detonators and protective molecules of AD. Finally, we demonstrated that total Tau and phosphorylated Tau protein were not modified by BGM dysbiosis, suggesting that Tau accumulation and phosphorylation could be regulated by other factors in the symptomatic stage of AD.

Finally, it is important to mention some limitations of this study. The Spearman analyses suggest interesting tendencies of correlation that did not reached statistical significance. Similar to other preclinical results, these should be interpreted with caution, as they may not translate directly to the human condition. Additionally, it is necessary to test out the functional metagenomic predictions observed to link multiple factors associated with BGM in AD onset and progression.

## 4. Methods and Materials

### 4.1. Animals

The study subjects were female (n = 16) and male (n = 8) triple-transgenic mice for AD (3xTg) harboring APP_Swe_ and Tau_P30L_ transgenes on a mutant PS1_M146V_ knock-in background, and female (n = 10) and male (n = 5) non-transgenic mice (NoTg) from the same genetic background B6129SF1/J (both Jackson Laboratory, Bar Harbor, ME, USA) at postnatal day 90 (PD90). All mice were housed in groups of 3–5 per cage with water and food (LabDiet 5001) *ad libitum* and maintained in a room with *12h/12h* light-dark cycle, lights on at 20:00 h. We performed all behavioral procedures between 9:00 and 13:00 h. We performed genotyping as previously reported [15] to confirm the 3xTg-AD mouse genotype.

### 4.2. Chronic Antibiotics Treatment

To evaluate the effect of manipulating BGM composition on memory impairment and the accumulation of Aβ and pTau in a preclinical model of AD, the female NoTg and 3xTg mice received an antibiotic cocktail consisting of ampicillin (1.0 g/L), vancomycin (0.5 g/L), metronidazole (1.0 g/L), and neomycin (1.0 g/L) in drinking water [49,50,86,87,88]. From PD90 to the end of the experiments, females were divided into groups: half of them, including NoTg and 3xTg mice, were allowed to drink water (Ctrl); the other half received the antibiotics (Abx). At PD100, females of every group were mated with male mice of the same genotype. The Ctrl or Abx treatments were administered to mothers (F0) during gestation and lactation of offspring (F1). At PD30, F1 was weaned, and male NoTg and 3xTg mice were selected for analysis. F1 mice continued on either Ctrl or Abx treatment until PD150. Liquid consumption was recorded in the first 30 days after weaning. Body weights were measured weekly, from PD1 to PD150 (Figure 9).

### 4.3. Fecal Samples Collection for Microbiota Analysis

At gestational days 18–19 (GD18–19), fresh fecal samples were collected in F0-mother NoTg and 3xTg-AD mice to evaluate the effect of Ctrl or Abx treatment over gut microbiota abundance or diversity during gestation. To observe BGM abundance, or diversity, vertically transmitted from F0 receiving Ctrl or Abx treatment, fecal samples of F1 NoTg and 3xTg mice were collected at PD30; with the same purposes, fecal samples of F1 were collected in the early symptomatic stage of AD, at PD150. Fresh fecal pellets were collected and stored until their DNA microbiota profile analysis, as previously reported [10].

### 4.4. DNA Extraction, Library Preparation, and 16S rRNA Gene Amplicon Sequencing

DNA extraction was carried out from 2–3 fecal pellets at the Laboratorio Estatal de Salud Pública del Estado de México, ISEM, Toluca de Lerdo, Estado de México. DNA extraction was carried out from 2–3 fecal pellets using the Zybio-EXM 3000 automatic nucleic acid extraction platform and nucleic acid extraction kit Cat. B-200-8 (Zybio Inc., Chongqing, China) (Cat: B200-8). The cartridges provided with the kit were centrifuged at 3000 rpm; subsequently, the fecal pellets from mice, 15 µL of proteinase K, and 500 µL of lysis buffer were added to each well. Once all the samples were loaded, the cartridges and rod covers were placed in the Zybio-EXM300^®^ instrument, and the Zybio B-200 program was used for DNA extraction. When the DNA extraction process was completed, DNA samples were aliquoted into microtubes and stored at −20 °C. DNA concentration was measured (260/280 abs) using a NanoDrop 2000 spectrophotometer (Thermo Scientific, Wilmington, MA, USA), and DNA quality was evaluated by electrophoresis in 0.5% agarose gel. The average concentration of samples was 87.82 ± 64.47 ng/μL. The amplification of the V3 hypervariable region of the 16S rRNA gene was amplified by PCR in a final volume of 50 μL, using the V3-341 reverse primer (0.2 μM) containing a different barcode per sample (1–100), and 1–10 ng of total DNA per reaction was used as a template. The information about Golay barcode, adapters, and primers utilized was reported in Corona-Cervantes et al. (2020) [55]. PCR master mix preparations and library preparation were accomplished according to Bello-Medina et al., 2021 [10]. In brief, the PCR condition was 95 °C for 5 min, followed by 30 cycles at 94 °C for 15 s, 62 °C for 15 s, and 72 °C for 15 s; and a 10 min extension at 72 °C using a GeneAmp PCR System 2700 Thermocycler (Applied Biosystems, San Francisco, CA, USA). For the library preparation, each of the 1–100 barcoded amplicons was quantified by gel densitometry and pooled. The mixture was purified using E-Gel iBase Power System (Invitrogen, Waltham, MA, USA), and the library’s size and concentration were checked using the Agilent 2100 Bioanalyzer system and High Sensitivity DNA Kit (Agilent Scientific Instruments, Santa Clara, CA, USA). High-throughput sequencing of the ∼281 bp single-end reads of the V3 region of the 16S rRNA genewere sequenced using Ion OneTouch ES 2, Ion PGM Template OT2 200 Kit v2 DL (Life Technologies, Carlsbad, CA, USA), Ion 318 Chip Kit v2, and Ion Torrent PGM System as previously described [89]. 

The sequence and corresponding mapping files for all samples used in this study were deposited in the NCBI BioSample repository (accession number: PRJNA000000). https://www.ncbi.nlm.nih.gov/sra/PRJNA000000 accessed on 25 November 2021.

### 4.5. Filtering and Taxonomic Classification 

Sequencing reads were filtered by the Torrent Suite™ Software 5.4: Life Technologies Corporation Carlsbad, CA, USA (PGM software) to remove low-quality and polyclonal sequences. Filtered data were analyzed using FastQC (v0.11.9) software and trimmed to 197 nucleotides using Trimmomatic (v0.38) software [90]; as a result, the demultiplexed sequence files (FASTQ) were obtained and then converted into a single concatenated FASTA file. The sequencing summary is shown in Appendix A. Demultiplexed sequences were used as input files in QIIME2 (Quantitative Insights into Microbial Ecology) v2.08 scripts [91]. DNA sequences were denoised and processed into amplicon sequence variants (ASVs) using DADA2 plugin. Taxonomy was assigned to ASVs using the q2-feature-classifier against-Green Genes database v13.8 [92]. 

### 4.6. Determination of Bacterial Abundance and Diversity

The Quantitative Insights into Microbial Ecology software (QIIME2) (v.2.08) [91] was used for the characterization of relative microbial abundance and beta diversity. For alpha-diversity analysis, indexes were determined using phyloseq [93] and plotted using ggplot2 packages in R environment (v4.0.3) (R Foundation for Statistical Computing, Vienna, Austria). For alpha diversity analyses, we used the Chao1 index (bacterial richness estimator) and the community diversity Simpson (dominance) and Shannon (evenness) indexes. In the case of beta diversity, the dissimilarity based on ASV abundance was estimated using weighted UniFrac distance metric to compare the bacterial community similarity, and three-dimensional scatter plots were generated using principal coordinate analysis (PCoA) with QIIME2 emperor. Significant differences in the relative abundance of bacterial taxa were detected by the Linear discriminant analysis effect size program (LEfSe) [94] using the Galaxy web platform (Afgan et al. 2016). The effect size of each taxon between all groups was estimated with LDA scores ≥ 2.5 and *p* > 0.05 [94].

### 4.7. Novel Object Localization Task

The apparatus used was an open acrylic box (33 cm × 33 cm × 33 cm) with black walls. The box was cleaned with 7% acetic acid (*v*/*v*). The floor of the box was covered with a 1 cm layer of sawdust that was replaced between sessions. In the walls of the experimental room, there were four visual cues within the visual field of mice. The objects used were rectangular boxes of 2.8 cm × 6.5 cm; these objects were called familiar object localization 1 (Fam1) and familiar object localization 2 (Fam2) or novel object localization (Nov). These objects were attached with Velcro^®^ to the box floor and were cleaned after each trial with 70% alcohol solution (*v*/*v*). A video camera was positioned above the box, and each trial was video recorded for post-training and post-retention analysis.

The F1 NoTg-Ctrl, NoTg-Abx, 3xTg-Ctrl, and 3xTg-Abx mice at PD150 were handled for 5 min every day for three consecutive days. The novel object localization task (NOL) consisted of three sessions: habituation, training, and retention tests. During the habituation, the mice could freely explore the open box, without objects, for 5 min; in this session, some motor activity parameters were evaluated such as speed, distance traveled, and resting time. Twenty-four hours after habituation, in the training session, the mice were placed in the open box, which contained the two sample objects (Fam1 and Fam2), for 10 min. The retention test took place 24 h after the training session. Mice were placed in the open box with one familiar object in the same place it had been during training (Fam), and the other familiar object in a novel location (Nov) (Figure 9). The results were expressed both as total exploration time per object and preference percentage of familiar object localization or novel object localization for training and retention test, respectively.

### 4.8. Immunohistochemistry for β-Amyloid and Tau

After the NOL retention test, the mice were anesthetized and euthanized (n = 4–5 mice per group); then, they were transcardially perfused with 4% paraformaldehyde in 0.1 M phosphate buffer via the ascending aorta. Brains were removed and post-fixed overnight in the same solution. They were cryoprotected with 30% sucrose in phosphate buffer for 5 days. Four frozen sagittal sections of 30 μm from the left hemisphere that contained the *subiculum*, CA1, and CA3 of the HIP (lateral with respect Bregma, from 0.72 to 1.80 mm [95]) were cut with a Leica cryostat and placed on slides. Immunohistochemistry was carried out as reported by Bello-Medina et al. (2021) [10]. The histopathological markers of AD used in this study were: total Aβ (anti-BAM-10, 1:500; Sigma-Aldrich, St Louis, MO, USA), Aβ1-42 (Aβ1-42, 1:500; ThermoFisher Scientific, Waltham, MA, USA), total Tau protein (Tau499, 1:500; donated by Dr. Wischik from the University of Aberdeen, UK), and phosphorylated Tau in Thr231 (pTau231, 1:500; Abcam, Cambridge, MA, USA). The secondary antibody used for each primary antibody was Alexa-Fluor 488 coupled to goat anti-mouse or anti-rabbit antibody (1:500; Life Technologies, Cambridge, MA, USA). The nuclei were counterstained with DAPI (1:5000; Sigma-Aldrich, St Louis, MO, USA). The stained sections were covered with fluorescence mounting medium (Fluoromount-G, Electron Microscopy Sciences, Hatfield, PA, USA).

The *subiculum*, CA1, and CA3 single mosaic images used in the analyses were obtained with a 40×/1.25 apochromatic objective lens, a filter set for Alexa 488 and DAPI detection, and the MosaiX module for the Apotome system (Zeiss, CDMX, Mexico). Analysis of the total Aβ, Aβ1-42, total Tau protein, and pTau231 area ratio in the *subiculum*, CA1, and CA3 was performed on the single optical plain images using ImageJ software. The procedure was carried out as described previously [10,96] (Appendix A). The results were expressed as the area ratio (area that is occupied for total Aβ, Aβ1-42, total Tau protein, and pTau231 in the total area of *subiculum*, CA1, and CA3 of the HIP of NoTg or 3xTg mice that received Ctrl or Abx treatment).

### 4.9. Correlation Analyses

Correlations between BGM composition (ASV abundances) and mouse metadata parameters of the NOL task (preference percentage of familiar object localization (training) or novel object localization (retention test)) and histological measure (area occupied for total Aβ or Aβ1-42 in each region of HIP) were calculated using Spearman rank correlation in R environment using the microbiome R package (R). Correlations between ASVs and host parameters were considered significant when *p* < 0.05. These correlations were calculated for F1 3xTg mice at PD150 including both treatment groups.

### 4.10. Metagenome Prediction of the BGM

We performed a computational approach to predict the functional composition of a metagenome from the 16S rRNA profile of BGM using Phylogenetic Investigation of Communities by Reconstruction of Unobserved States (PICRUSt) [97].

### 4.11. Statistical Analysis

The consumption of Ctrl or Abx treatments in F1 NoTg or 3xTg mice at PD30 to PD60 was estimated as the mean of liquid consumption by all mice that were housed in the same home-cage. We performed the Kolmogorov–Smirnov test to prove the data normality parametric assumption. For body weight of F1 NoTg or 3xTg mice, which was registered for 23 weeks, a repeated measures three-way ANOVA was used, where factor 1 was genotype (NoTg or 3xTg), factor 2 was treatment (Ctrl or Abx), and factor 3 was age (23 weeks). The motor activity data analysis was performed with a two-way ANOVA, where factor 1 was genotype (NoTg or 3xTg) and factor 2 was treatment (Ctrl or Abx). The NOL performance analysis with exploration time measures was made with a three-way ANOVA, where factor 1 was genotype (NoTg or 3xTg), factor 2 was treatment (Ctrl or Abx), and within-subject factor 3 was object localization (Fam1 vs. Fam2 in training or Fam vs. Nov in retention test). For preference percentage of familiar object localization (training) or NOL (retention test) a two-way ANOVA was performed, where factor 1 was genotype (NoTg or 3xTg) and factor 2 was treatment (Ctrl or Abx).

For the total Aβ, Aβ1-42, total Tau protein, and pTau231 immunohistochemical analysis, we applied a two-way ANOVA, where factor 1 was genotype (NoTg or 3xTg) and factor 2 was treatment (Ctrl or Abx) for each HIP area (*Subiculum*, CA1 or CA3). We used the post hoc Bonferroni when appropriate. A *p*-value < 0.05 was considered statistically significant. 

Statistical analyses of BGM data were performed using SPSS statistics V 28.0.0 (IBM, New York, NY, USA) to compare the alpha-diversity indexes among groups, and GraphPad Prism software v 9.0 to compare relative abundance across groups. Non-parametric Mann–Whitney U test was used to calculate the significance between two groups, considering a *p*-value < 0.05 as statistically significant. A PERMANOVA analysis was used for category comparison of distance matrices (UniFrac). Behavioral and histological data were analyzed for association with bacterial relative abundance, and bacterial diversity using Spearman and Pearson correlation. Pearson correlation analysis was performed in GraphPad Prism v.9.0.0 (San Diego, CA, USA) to identify the correlation of bacterial diversity (Shannon index) with the measures of behavioral and histological data. *p* < 0.05 was considered significant.

To determine significant differences in BGM abundance and metabolic pathways, Statistical Analysis of Taxonomic and Function software (STAMP, v2.1.3) was used. A Kruskal–Wallis H test and Tukey-Kramer post hoc test with Benjamini-Hochberg FDR multiple test correction were used to estimate the false discovery rate (FDR) and filter the data considering a *q*-value < 0.05.

## Figures and Tables

**Figure 1 ijms-23-08209-f001:**
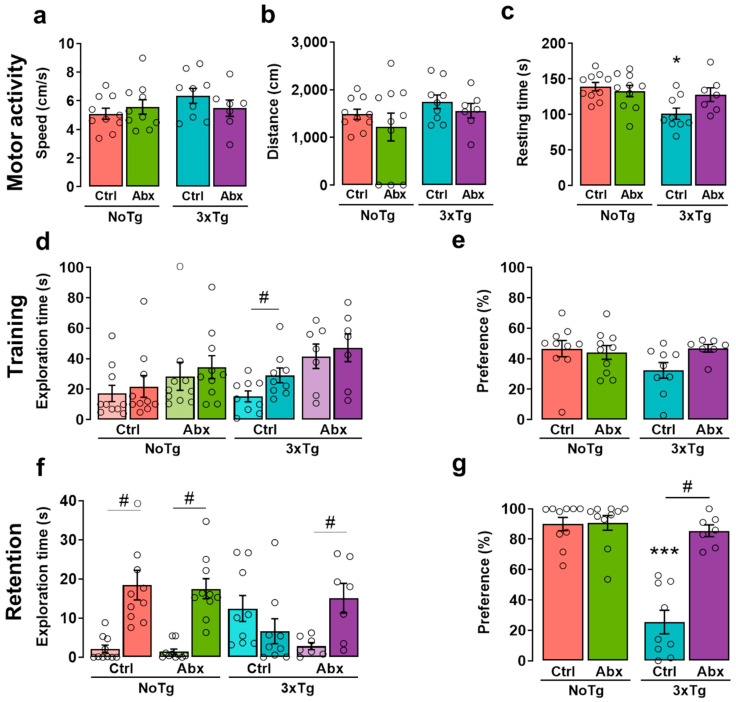
BGM dysbiosis with Abx treatment slows the progression of motor and memory impairment at the early symptomatic stage of AD. Behavioral results in the novel object localization (NOL) task. Mean (±SEM) of (**a**) speed, (**b**) distance traveled, and (**c**) resting time measured in motor activity during the NOL habituation session. (**d**) Exploration time of familiar object localization 1 (Fam1) in light-colored bars and familiar object localization 2 in dark-colored bars (Fam2; three-way ANOVA) and (**e**) preference percentage of familiar object localization in the training session. (**f**) Exploration time of familiar object localization (Fam) in light-colored bars and novel object localization (Nov) in dark-colored bars and (**g**) preference percentage of novel object localization in the retention session of offspring (F1) male non-transgenic (NoTg) or Alzheimer’s disease triple-transgenic (3xTg) mice that received water (Ctrl) or antibiotics (Abx) from gestation to PD150. The results were analyzed with a two-way ANOVA for a, b, c, e, and g data or a three-way ANOVA for d and f data. * *p* < 0.05, *** *p* < 0.0001 vs. NoTg group; # *p* < 0.05 to <0.0001 between groups; n = 7–10 mice per group. In this Figure and in the following Figures, the small circles represent the data of individual animals.

**Figure 2 ijms-23-08209-f002:**
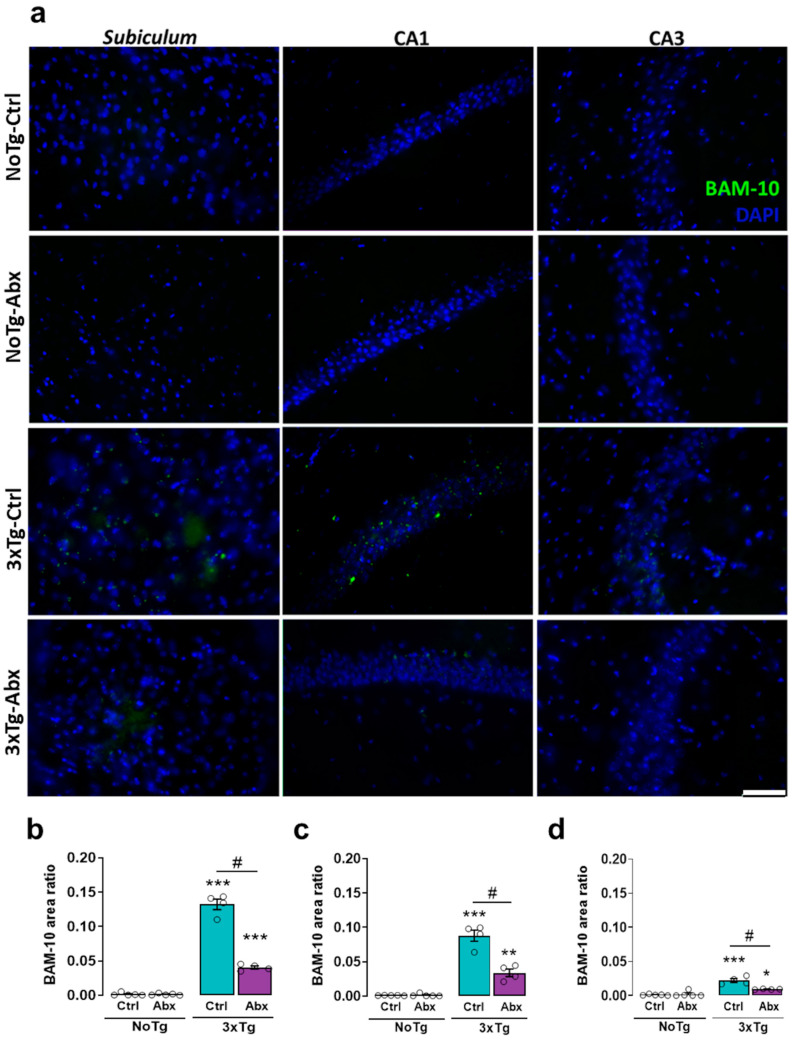
BGM dysbiosisdysbiosis with Abx treatment slows the accumulation of total Aβ (BAM-10) in the hippocampus at the early symptomatic stage of AD. Immunohistochemistry analysis for total Aβ. (**a**) Representative images show BAM-10 (green) and DAPI nuclei detection (blue) acquired from the hippocampus with a 40× objective. Mean BAM-10 area ratio (±SEM) in the (**b**) *subiculum*, (**c**) CA1, and (**d**) CA3 of F1 male non-transgenic (NoTg) or Alzheimer’s disease triple-transgenic (3xTg) mice that received water (Ctrl) or antibiotics (Abx), from gestation to PD150. The results were analyzed with a two-way ANOVA, * *p* < 0.05, ** *p* < 0.01, *** *p* < 0.0001 vs. NoTg group; # *p* < 0.05 to <0.0001 between groups; n = 4–5 mice per group. Scale bar = 20 µm.

**Figure 3 ijms-23-08209-f003:**
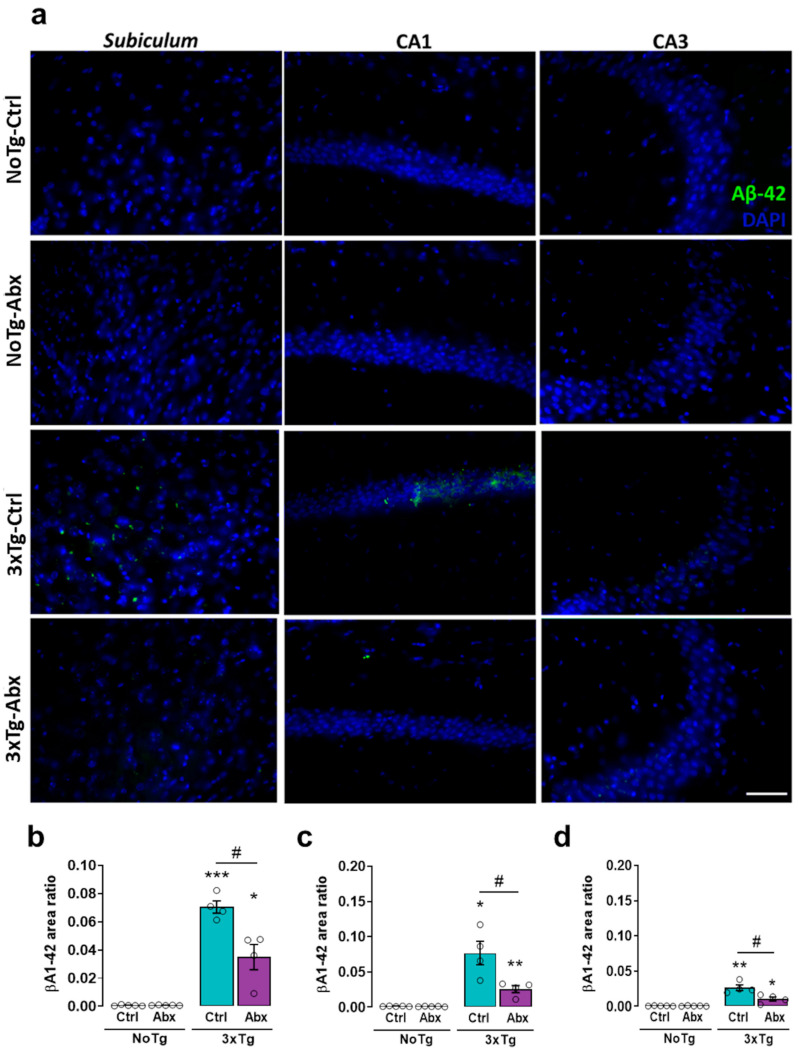
BGM dysbiosisdysbiosis with an antibiotics (Abx) treatment slows the accumulation of Aβ1-42 in the hippocampus at the early symptomatic stage of AD. Immunohistochemistry analysis for Aβ1-42 (**a**) Representative images show Aβ1-42 (green) and DAPI nuclei detection (blue) obtained with a 40x objective in the hippocampus. Mean Aβ1-42 area ratio (±SEM) in the (**b**) *subiculum*, (**c**) CA1, and (**d**) CA3 of F1 male non-transgenic (NoTg) or Alzheimer’s disease triple-transgenic (3xTg) mice that received water (Ctrl) or antibiotics (Abx) from gestation to PD150. The results were analyzed with a two-way ANOVA, * *p* < 0.05, ** *p* < 0.01, *** *p* < 0.0001 vs. NoTg; # *p* < 0.05 to <0.0001 between groups; n = 4–5 mice per group. Scale bar = 20 µm.

**Figure 4 ijms-23-08209-f004:**
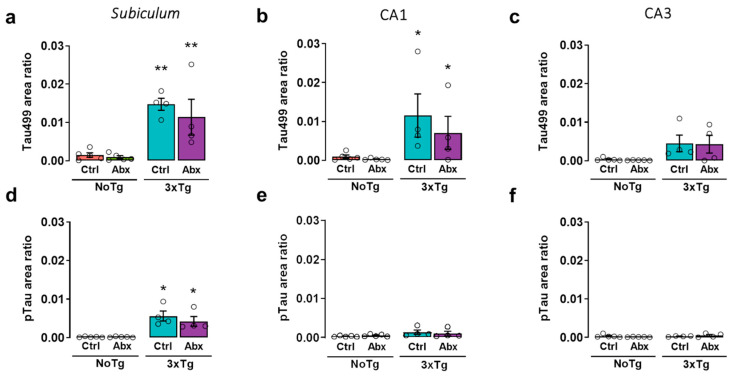
BGM dysbiosis with Abx treatment does not change the accumulation of total Tau (Tau499) and phosphorylated Tau (pTau) in the hippocampus at the early symptomatic stage of AD. Immunohistochemistry analysis for Tau499 and pTau. Mean Tau499 area ratio (±SEM) in (**a**) *subiculum*, (**b**) CA1, and (**c**) CA3 and mean pTau area ratio (±SEM) in (**d**). *subiculum*, (**e**) CA1, and (**f**) CA3 of F1 male non-transgenic (NoTg) or Alzheimer’s disease triple-transgenic (3xTg) mice that received water (Ctrl) or antibiotics (Abx) from gestation to PD150. The results were analyzed with a two-way ANOVA, * *p* < 0.05, ** *p* < 0.01 vs. NoTg; n = 4–5 mice per group.

**Figure 5 ijms-23-08209-f005:**
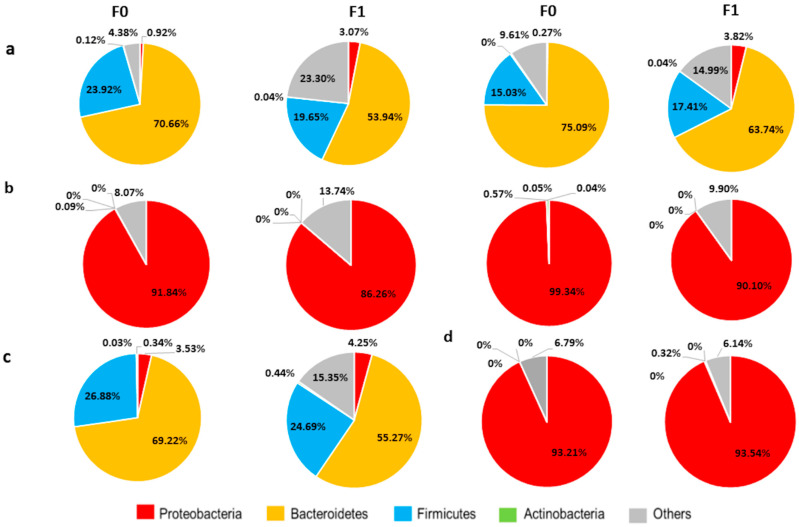
Profile comparison of relative abundance at phylum level in (F0) mothers and (F1) offspring treated or not with antibiotics. Stool samples from F0 were obtained on gestational day 18–19 and on postnatal day 30 for F1. (**a**) Representative data from two non-transgenic mothers without antibiotics treatment and their respective F1 (in pie chart in front) (**b**) Profile of two non-transgenic mothers and their offspring that received the antibiotics. (**c**) Alzheimer’s disease triple-transgenic mouse with its respective F1. (**d**) Profile of F0 3xTg-Abx with their respective F1. Each pie chart represents one animal.

**Figure 6 ijms-23-08209-f006:**
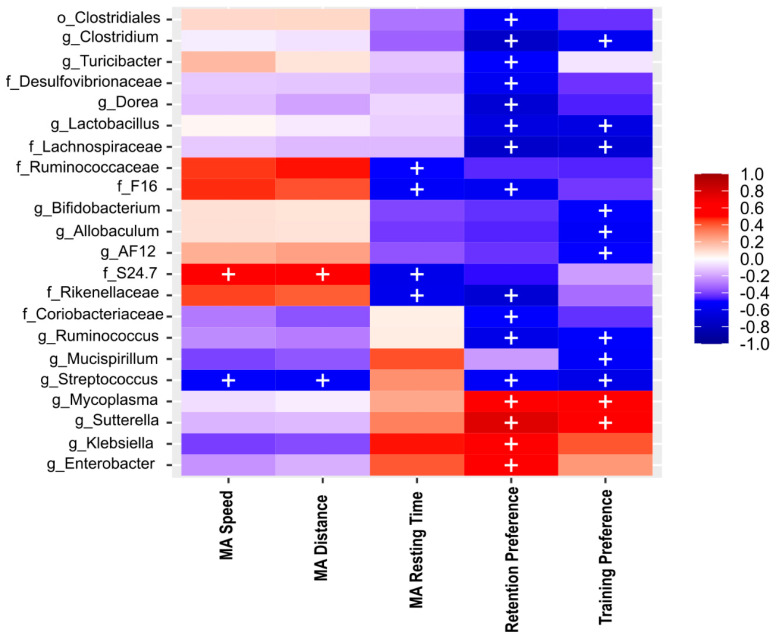
Heat map of Spearman correlation between the relative abundance of the main phyla and 22 genera (see rows) identified in BGM and behavioral performance in motor activity (MA), such as speed (MA speed), traveled distance (MA distance), and resting time (MA resting time), also the preference percentage of familiar object localization in training and novel object localization in retention test of NOL (see columns) in Alzheimer’s disease triple-transgenic (3xTg) mice that received water (Ctrl) or an antibiotics (Abx). The heat colors indicate positive correlations, and cool colors indicate a negative correlation. + symbol represents significant differences (*p* < 0.05) observed in the correlation analysis.

**Figure 7 ijms-23-08209-f007:**
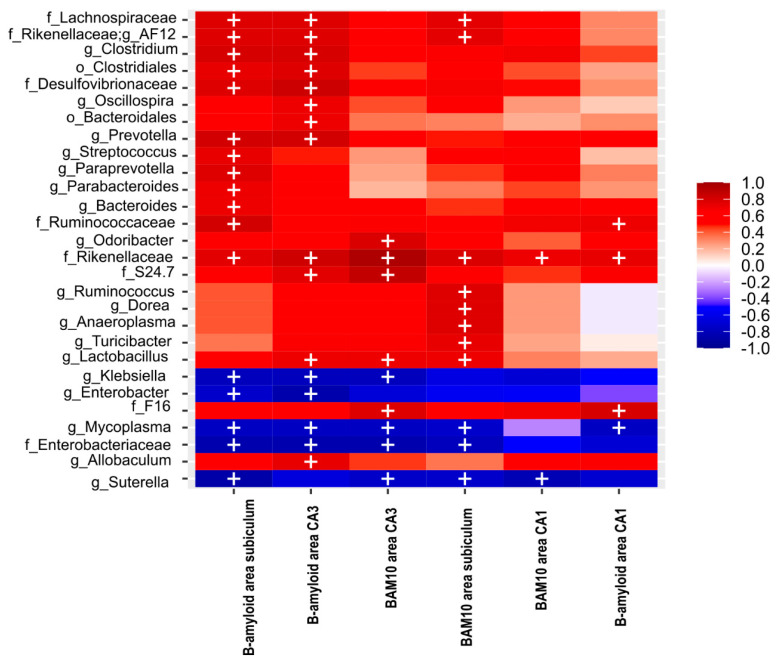
Heat map of Spearman correlation between the relative abundance of the seven main phyla and 22 genera (see rows) identified in BGM microbiota and the total Aβ (BAM10) area ratio in the *subiculum*, CA1, and CA3 of the HIP, as well as the Aβ (B-amyloid) 1-42 area ratio in the *subiculum*, CA1, and CA3 of the HIP (see columns) in Alzheimer’s disease triple-transgenic mice (3xTg) that received water (Ctrl) or an antibiotics (Abx). The heat colors indicate positive correlations, and cool colors indicate a negative correlation. + symbol represents significant differences observed in the correlation analysis.

**Figure 8 ijms-23-08209-f008:**
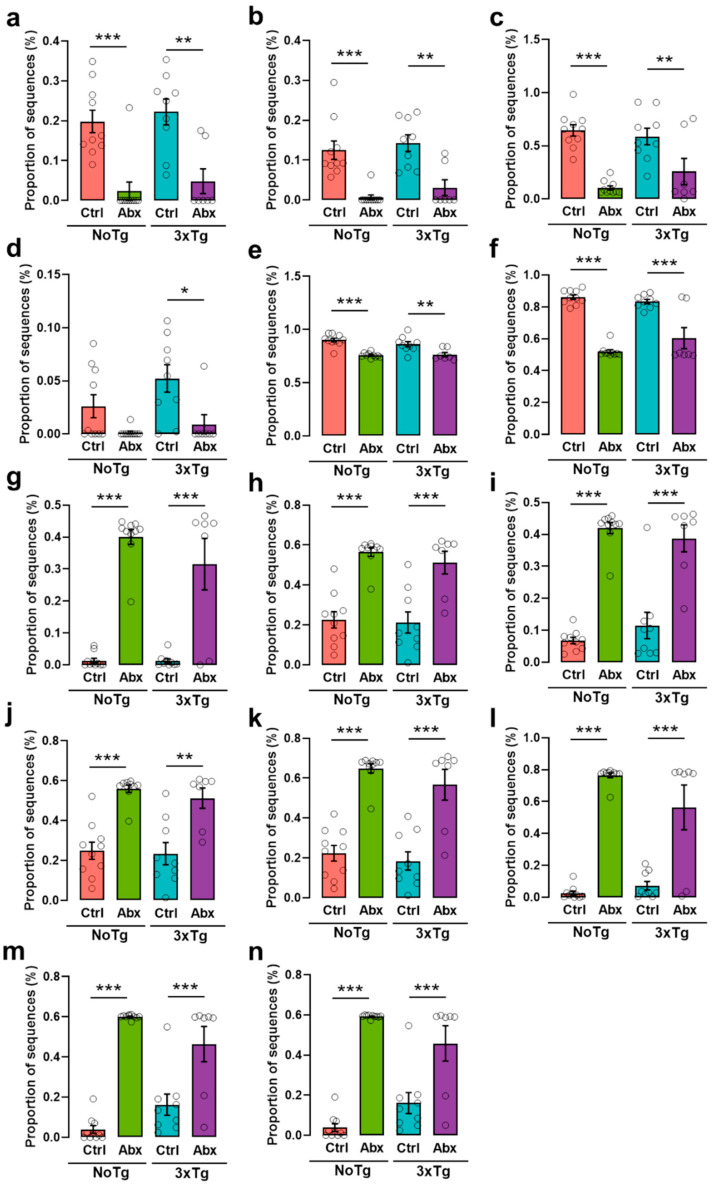
BGM dysbiosis with an Abx reduces the metabolism of detonator molecules such as (**a**) acetate, (**b**) acetate and butyrate, (**c**) propanoate, (**d**) butyrate, (**e**) diacylglycerol, and (**f**) peptidoglycan; and greater metabolism of protective molecules such as (**g**,**h**) biotin precursors, (**i**) biotin cofactor, (**j**) vitamin B, (**k**) oleate, (**l**) alanine, (**m**) phenylalanine, and (**n**) tyrosine in F1 male non-transgenic (NoTg) or Alzheimer’s disease triple-transgenic (3xTg) mice that received water (Ctrl) or an antibiotics (Abx) from gestation to PD150. All graphs show the mean proportion of sequences (±SEM). * *p* < 0.05, ** *p <* 0.01, *** *p* < 0.0001 vs. NoTg or 3xTg; n = 7–10 mice per group.

**Figure 9 ijms-23-08209-f009:**
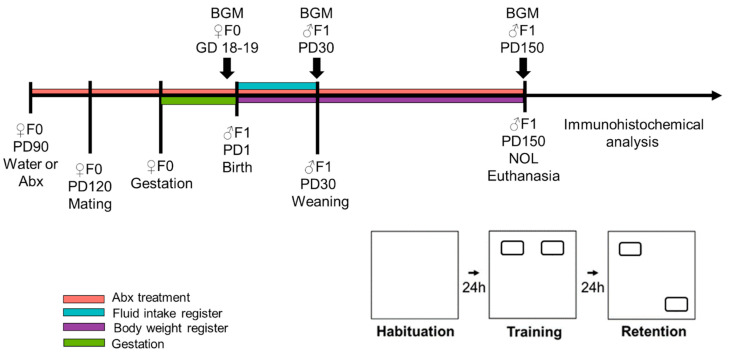
Schematic representation of the experimental design. Female non-transgenic (NoTg) or Alzheimer’s disease triple-transgenic (3xTg) mice constituted F0. F0 received water (Ctrl) or an antibiotics treatment (Abx) from postnatal day (PD) 90; one month later they were placed mated. Antibiotic treatment or water only continued during gestation and lactation. Weaning of F1 offspring occurred at PD30. F1 males received the same treatment as F0 but from gestational day (GD) 18–19 to PD150. Body weights were measured during this time. Fluid consumption was measured during the first 30 days of treatment. The novel object localization (NOL) task was performed on PD150. NOL consists of three sessions: habituation, training, and retention tests. After the NOL task, the mice were immediately euthanized, and their brains were processed for β-amyloid and Tau immunohistochemistry analysis in the HIP. Fecal samples were collected in F0 at GD18–19, in F1 at PD30 and PD150 for 16S rDNA gene sequencing analyses of bacterial gut microbiota (BGM).

## Data Availability

The datasets generated for this study are available on request to the corresponding author.

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
