# Peer review of "Chronic-Antibiotics Induced Gut Microbiota Dysbiosis Rescues Memory Impairment and Reduces β-Amyloid Aggregation in a Preclinical Alzheimer’s Disease Model"

_ijms, 2022, doi:10.3390/ijms23158209_

Round 1

Reviewer 1 Report

Alterations in the bacterial gut microbiota (BGM) in patients with Alzheimer’s disease have been reported a lot. In this study, the authors specifically investigated BGM alteration related spatial memory impairment and corresponding Aβ- and Tau-associated molecular mechanisms at the early-symptomatic stage using 3xTg AD mouse model. The data are interesting, but there are some comments that the author should address.

1)      According to figure 6, vertical transfer of BGM from F0 to F1 occurred in both control non-transgenic mice and 3xTg AD mice; according to figure 9, BGM perturbation with an Abx reduces the metabolism of detonators molecules in both control non-transgenic mice and 3xTg AD mice, with nearly the same changing trend. These results appear to suggest that BGM change is unrelated to Aβ and Tau pathology, which contradicts the data in figures 3,4,5. Could the author elaborate on this?

2)      The presentation of data should be improved. For example, there is no Figure 1, and the authors need to re-arrange the figures in the main text.

3)  Figure S1a, no error bar was shown.  How was the data analyzed? Is the liquid consumption value an average number?

4) Figure 1d,f,g, what is the meaning of “#”?  Please keep consistent with the figure legends.

5) Please specify group Fam1, Fam2, Fam, Nov in Figure 1d,f.

6) Why are there some small gray dots near the red, yellow, green and gray rectangle symbols in supplementary figure 3?

7) Statistic analysis is confusing, the authors seemed to use “#” in some figures (Fig.2, Fig. 3), but use “*” in other figures (Fig.4).

8) In Fig. 5, the “*” symbol should be moved above the circle symbols.

9) Line 156, “(Fig. 3a and d)” should be “(Fig. 4a and d)”

10) Figure 6a, 6b, please specify each pie graph. Any differences between the two F0 and two F1?

Author Response

We appreciate the reviewer's suggestion.

The manuscript was reviewed. Please see the attachment.

Reviewer 2 Report

In this manuscript, Bello-Medina et al aim to demonstrate that antibiotic cocktail treatment in the Alzheimer’s disease mouse model 3xTg (and corresponding wild-type) has profound changes to the bacterial gut microbiota (BGM) which can be transferred to the F1 generation. This alteration in the BGM leads to slowing down of the progression of amyloid pathology in the transgenic mice, which may be prevent memory deficits. Importantly, they
show that this BGM alteration has no effect on the Tau pathlogy observed in these mice. Next, they perform correlational studies to look the bacterial abundance and diversity with the pathological features described earlier. Finally, they performed bioinformatic analyses of the bacterial metagenome to identify how the metabolic pathways and thereby production of certain metabolic molecules could be affected by the BGM alteration and draw inferences
about how this might relate to inflammation and AD pathology.

While the premise of the paper is intriguing, a major problem with the research design is that the antibiotic cocktail is continued for the entire duration of the experiment even in the F1 mice. This leads to the question whether the altered BGM is a result of inheritance from the antibiotic-treated dam or whether this particular cocktail pressure results in the observed microflora. To truly establish the role of vertical inheritance, and more importantly
maintenance, the F1 antibiotic treatment should have been stopped after weaning to see whether their BGM is altered for good and whether they show change in pathological progression. Further, would the results have been any different if just the F1 were treated with the same cocktail post-weaning to P150?

Moreover, it does not appear that the BGMs are only slightly different between non-treated Tg and non-transgenic mice, and nor is it different between the treated Tg and non-transgenics (Supp Fig 3 and Fig 6). Considering the way crosses were set-up where Tg mice and non-transgenic mice came from different parents and were never co-housed,
is it possible that the few differences that they authors found were arising from vertical inheritance by the F0 of a microbiome drift that may have occurred across the parent cages?

I have some additional concerns that I list below:

1.    The extended period of reduced fluid consumption in the 3xTg-Abx (Supp Fig 1) and reduced resting time in Fig 2C suggest that 3xTg might display some anxiety-like behavior which has been reported before in this line. Thus, instead of a subtle motor alteration, the differences could simply arise from anxious behavior. Have the authors tested this?

2.    The authors should include the color scheme for their NOL objects in Fig 2D&F. 

3.    While the authors mention that they have performed Bonferroni corrections for their multiple comparison analyses, the p-values reported often seem to be too high to pass the Bonferroni cut-off. For example, Fig 2D Fam1 vs Fam2 in 3xTg-NoAbx : p = 0.04. For a three-way ANOVA with so many potential group comparisons, the Bonferroni correction would be severe and the threshold for p value would be extremely small (0.05/C(8,2). Even if limited tests were chosen, even then a p value of 0.04 won’t be significant with a Bonferrnoni correction. The authors should check their statistics again. The same holds true for ln128-129, and in subsequent figures.

4.    For Fig 3-5, the authors have done their measurements on only one section per animal. This analysis is prone to errors and heavily influenced by the choice of the section. The authors should take systematic sampling of sections from different parts of the hippocampus along the AP axis and make these measurements.

5.    For Fig 3-5, the authors should express the results as fraction of area covered (area covered by antibody signal/total area of structure) instead of just area covered. This is particularly important when working with sections that have not been matched for size and position of brain structures being evaluated.

6.    The authors should also use an alternative method to validate the amelioration of the amyloid and Tau pathologies, as quantitative immunohistochemistry is often sensitive to staining, imaging and analysis conditions/settings.

7.    Fig 7 and 8 – The authors have run correlational analysis of different bacterial species against behavioral and neuropathological features. The authors have done no multiple comparison corrections for the p-values, nor have they regressed out dependent variables. Many of the behavioral features are co-varying, and are directly dependent on neuropathological state of the brain (i.e. amyloid load, tau load). Thus, it is not surprising that several genera show significant positive or negative correlations with several of the variables tested.

While it makes sense to perform such analysis for the amyloid and tau levels as there is a link between BGM alteration and change of neuroinflammatory status of the brain and thereby amyloid pathology, the behavioral deficits and recovery depend on the amyloid levels and are not be directly dependent on the BGM. The authors should move Fig 7 to the supplementary section. Also, the number of genera tested in Fig 7 and 8 are different. Can the authors explain this discrepancy?

8.    The discussion is running too long and the authors should consider shortening it. For example, references that support some of the bacterial genera differences observed in the study may be moved to the results. Information is repeated in the current form.

9.    While there are grammatical errors throughout the manuscript, they are particularly jarring in the discussion section making it difficult for the reader to understand the point that authors are making. The authors may consider using language editor services to improve the manuscript’s readability.

Author Response

(The authors gave the same response as above.)

Round 2

Reviewer 1 Report

The authors adequately addressed my concerns. No more comments!